# Oil/Water Separation Using Waste-Derived Functional Materials with Special Wetting Behavior

Arun K. Singh 

Department of Chemistry, M. M. Engineering College, Maharishi Markandeshwar (Deemed to be University), Mullana, Ambala 133207, Haryana, India; aruniitr09@gmail.com

**Abstract:** The separation of both emulsified and immiscible oil/water mixtures using materials with special wetting behavior (hydrophobic-oleophilic or hydrophilic-oleophobic and switchable wettability) has attracted significant research attention in recent years. Among various materials with special wetting behavior, waste material-mediated surfaces have gained more interest because of their various advantages such as fluorine-free and specific surface structural properties, vast availability, flexibility in chemical/structural modification to improve the desired surface properties and eco-friendly nature. This review article describes the oil/water separation application by the waste materials-based special wettable surfaces from different resources. The special wettable surfaces preparation method, treatment effect, efficiency and oil/water separation mechanism are discussed. Moreover, unresolved issues and potential challenges associated with all waste-derived special wettable materials have been highlighted for future development.

**Keywords:** special wetting behavior; waste-derived materials; oil/water separation; underwater superoleophobicity; underoil superhydrophobicity



## 1. Introduction

The oil spill accident and frequent discharge of oily contaminated water including organic effluents (such as toluene, hexane, xylene, benzene and dichloromethane) from various petroleum and food industrial processes without pre-treatment or purification not only lead to economic loses but also pose a serious threat to the terrestrial ecosystem and human health [1–8]. Additionally, marine animals and seabirds are also highly affected by the leakage of oils, organic solvents and its derivatives into the sea because of their floating ability on the surface of water [2,9]. Therefore, considering the environmental and economic concern, in the recent past, several methods such as chemical oxidation, centrifugation, biological treatment, flotation, electrochemical methods and adsorption have been reported for the treatment of oily wastewater [10–17]. Among them, materials with special wetting behavior qualities (superhydrophobic-superoleophilic or superhydrophilic–superoleophobic or superhydrophilic and underwater superoleophobic) have drawn considerable attention of researchers for oil/water separation via filtration or adsorption processes [6,18,19].

Metallic nanoparticles are highly advantageous in various industrial and daily life applications such as water remediation, antiviral, antimicrobial, agricultural applications including the development of materials with special wetting behavior [20–25]. Materials with special wetting behavior tend to have selective liquid repellence ability (either oil or water phase at a time) on their solid surfaces [5,15,26].

Recently, many synthetic materials with special wettability such as metallic mesh [17,27–31], cotton fabrics [3,32,33], membranes [34–38], glass filters [39,40] and sponges [41,42] have been developed by coating of specific low surface energy chemicals along with the incorporation of metallic nanoparticles and organic compounds and applied for the treatment of oily and organic solvent contaminated water [43,44]. However, these materials are associated with some concerns and defects, such as high preparation cost, complicated

preparation process, utilization of fluorinated compounds and are environmentally un-desirable, which limits their application for the treatment of oily contaminated water in real field contexts [45–47]. Additionally, only a few can separate oil-in-water or water-in-oil emulsion which is stabilized by particular surfactants [46,48]. Therefore, in order to overcome the issues discussed above, several researchers have reported the use of various biodegradable agro-industrial waste materials as a promising substitutes for synthetic materials with special wettability for efficient oil/water separation using eco-friendly approaches [1,49–51]. The recycling of waste materials in order to develop new and useful resources for the treatment of oily contaminated water is of great value not only for the fabrication of extremely low cost special wettable surfaces but also become a significant reason to minimize environmental pollution [6,26]. In the recent past, numerous waste materials from different resources have been investigated oil/water separator because of their special wetting behavior with respect to oil and water [1,5,26,52]. From the viewpoint of the United Nations' Sustainable Development Goals (SDGs), the recycling of biomass and industrial waste materials to develop green and eco-technologies for water treatment can help to attain SDGs 6 (clean water and sanitation) [53–56].

In this sense, the current review is focused on the in recent progress in the employment of agro-industrial waste materials with switchable wettability for oil/water separation application. The utilization of waste materials from different sources is systematically discussed on the recovery or separation of oily and organic solvents from contaminated water along with their conceptual mechanisms. Additionally, significant factors influencing the performance of oil/water separation, associated challenges and essential improvements are also summarized.

## 2. Waste Materials with Special Wettability for Oil/Water Separation

Natural and lignocellulosic materials such as fruit peels, peanut shell, potato peels, walnut shell, coconut shell, saw dust, waste paper and cigarette filter are easily available, readily biodegradable and non-toxic in nature [5,26,52,57–62]. The upsides of these materials having high cellulosic content with low bulk density, fluffiness and hydrophilicity, owing to the presence of abundant hydroxyl (–OH) groups. Because of these specific properties, such materials are highly useful in the fabrication of underwater superoleophobic or under oil superhydrophobic as a promising filter for the separation of oil/water mixtures [58,59]. In addition, some other waste materials such as waste plastic, polyethylene gloves, coal fly ash and candle soot are also useful as coating materials for the development of materials with special wetting behavior (superhydrophobic–superoleophilic) by lowering the surface energy and increasing the roughness of the target substrates. Thus, superhydrophobic–superoleophilic, underwater superoleophobic–superhydrophilic, underoil superhydrophobic–superoleophilic and switchable wettability surfaces can be prepared by the use of agro-industrial waste materials in a specific optimized ratio and amount (Figure 1). In the next section, fabrication of various types superhydrophobic–superoleophilic, underwater superoleophobic–superhydrophilic and underoil superhydrophobic–superoleophilic surfaces using waste materials are discussed along with their oil/water separation performance and respective mechanisms.

### 2.1. Underwater Superoleophobic-Superhydrophilic Surfaces

Zhao et al. (2020) [57] reported the fabrication of oil/water separation layer with under water superoleophobic ability by the use of waste peanut shell. In this work, the authors smashed the peanut shells (previously washed with deionized water and dried at 80 °C for 6 h) with the help of a grinder in order to obtain in the form of powder with particle size in the range from 34.7 to 425 μm. They developed oil/water separation layers by the accumulation of peanut shell powders between the nylon meshes (as shown in Figure 2a). The wettability of the modified peanut shell powder was examined with respect to oil and water. They observed that when a droplet of oil comes into contact with the peanut shell powder surface in air, oil is absorbed, but under water the developed peanut

shell powder rejects the oil with an oil contact angle of 145–160°, revealing the underwater superoleophobicity due to the presence of a hierarchical porous structure and chemical composition of peanut shell such as cellulose, hemicellulose and lignin being responsible for hydrophilicity.

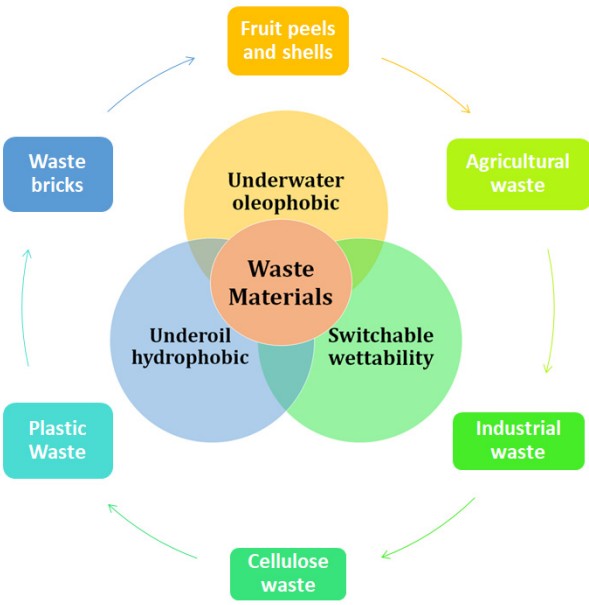

**Figure 1.** Schematic illustration agro-industrial waste-derived materials which can be used to develop surfaces with special wetting behavior.

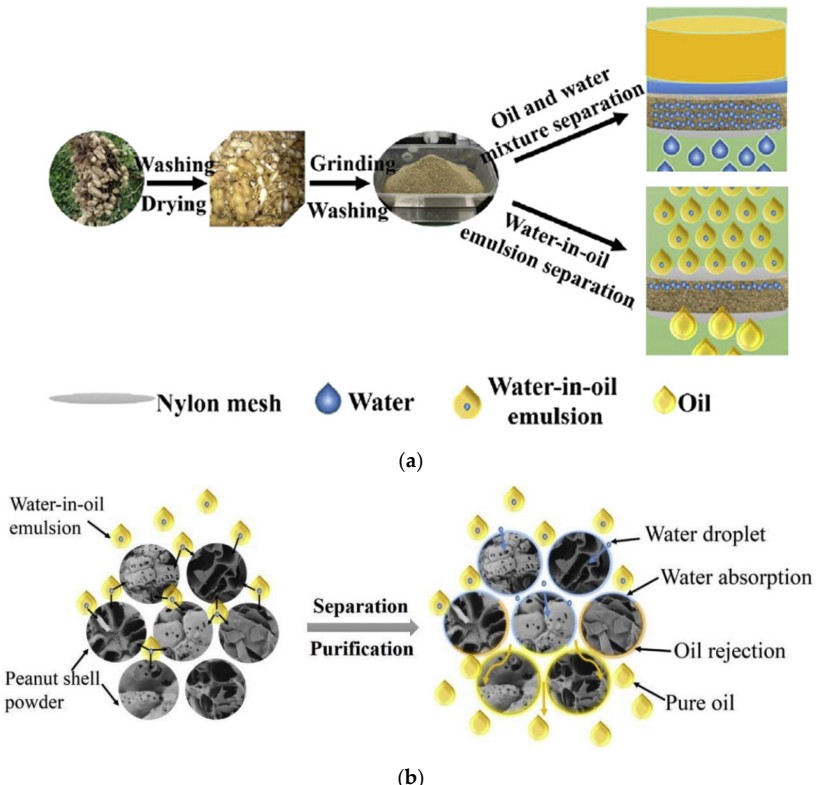

**Figure 2.** (**a**) Schematic representation of preparation of peanut shell-based nylon mesh for oil water separation, (**b**) Schematic representation of mechanism of water-in-oil emulsion separation through the underwater superoleophilic layer of peanut shell powder. Reproduced with permission from [57]. Copyrights (2020), Elsevier.

Because of this selective wetting behavior with respect to oil and water, authors examined oil/water separation efficiency of the developed layer in underwater conditions via continuous filtration approach. The authors observed very high oil/water separation efficacy (>99%) with respect to various mixtures of water with oils or organic solvents including diesel oil, soyabean oil, petroleum ether, hexane and toluene even after various repeating (>50) cycles. In addition to the separation of immiscible oil/water mixture, the developed layer was also highly efficeint (>99%) in separation of water-in-oil emulsion prepared with kerosene and diesel oil. This behavior might be due to the smaller size of pore radius of separation layer as compared to the radius of emulsion droplet as well as strong water-capturing capability of hemicellulose and cellulose of peanut shell layer. Authors reported that water droplets absorbed on the powdery surface of superhydrophilic peanut shell and trapped water coalesced in the porous structure because of the capillary forces and generated water-solid interfaces. Thus, this reduces the interaction of oils with powder surface and continuously repelled oil from the surface of separation layer by the demulsification process [Figure 2b]. Thus, the authors concluded that that fabrication of a separation layer with the help of agricultural waste plays a significant role in the separation process of oil/water mixtures and emulsions.

Gan et al. [63] developed an aerogel-based membrane from the plastic waste combining a solvent replacement and freeze-drying route for the separation of surfactant-stabilized oil-in-water emulsions. The authors fabricated the aerogel membrane with the help of aramid nanofiber aerogel, kelvar, potassium hydroxide, dimethyl sulfoxide and tertbutyl alcohol. Additionally, they prepared aramid in 10 g of KOH and kelvar in 400 mL DMSO solution and stirred it for at least 5 days at 25 °C. Furthermore, the aramid nanofiber was added into tertbutyl alcohol solution for the protonation of aramid nanofiber for the formation of cross-linkage networks. In the second step, the freeze-drying method involved for the modification of aramid nanofiber aerogel membrane was used (Figure 3). The authors coated aramid nanofiber hydrogel manually on glass substrate and left it to freeze in liquid nitrogen for 5–10 min, and then placed it into the freeze-drying device for the development in a three-dimensional network for 48 h at −50 °C. The authors examined the oil/water separation ability of the modified membrane using Tween80-stabilized emulsions via gravity driven method. They observed that after pouring the surfactant-stabilized emulsions over the fabricated membrane, water passed easily and oil droplets remained above switching the emulsion from the turbid to clear/transparent solution revealing the underwater superoleophobicity and hydrophilicity by the membrane. The author described the mechanism behind the separation of oil/water emulsion based upon the "size sieving" effect and tiny droplets which passed through the filter layer colliding to form large droplets due to the generation of tortuous micro-channels inside the aerogel membrane.

Authors also examined the contact angle and oil/water separation property of the prepared membrane and reported that the developed material have a 147° oil contact angle and 98% separation efficiency. On the basis of experimental findings, the authors concluded that the utilization of plastic waste was highly effective in the development of underwater super oleophobic surfaces for the oil/water separation application.

Tian et al. [64] used waste aluminum sludge for developing a multifunctional layer (under water superoleophobic) by hydrothermal and calcination methods using hexamethylenetetramine, potassium dihydrogen phosphate and magnesium nitrate hexahydrate as modifying agents. The authors prepared layers of double oxides using waste aluminum sludge via calcination, grinding and sieving processes. Initially, they smashed waste aluminum sludge into fine granules (particle size ranging from 150 to 300 μm), and after performed calcination at 500 °C. Additionally, hydrothermal treatment was given to the layered double hydroxide for the fabrication of layered double oxide nanosheets on the treated waste aluminum sludge surface. Authors added 2.24 g of hexamethylenetetramine, 4.1 g of Mg $(NO_3)_2$ and 2 g in 120 mL deionized water and heated it at 120° for 12 h for

the formation of Mg–Al-based layered double hydroxide nanosheets and sandwiched then between two stainless-steel mesh wire meshes (as shown in Figure 4).

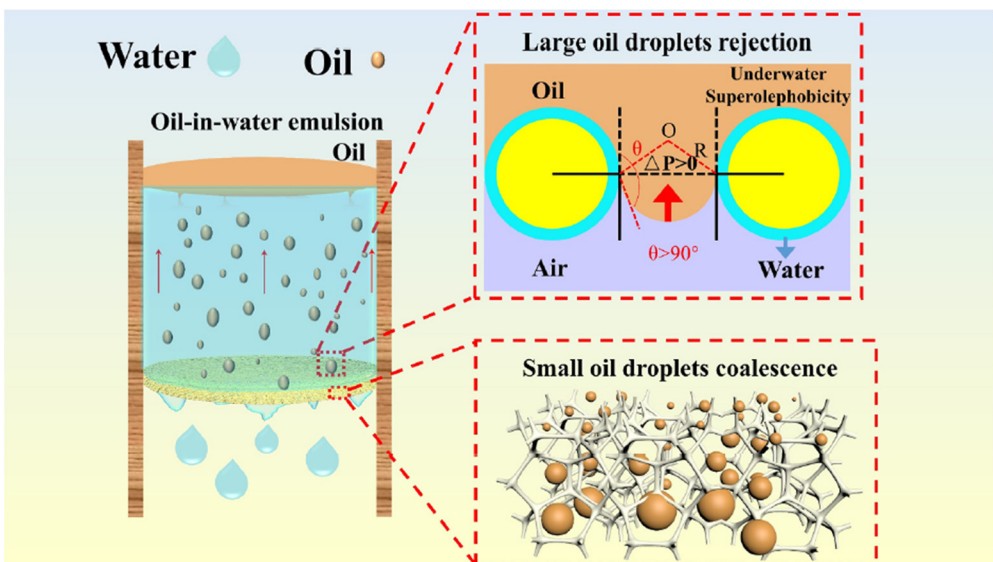

**Figure 3.** Schematic presentation of mechanism for separation of oil/water emulsion using aerogel-based membrane derived from plastic waste. Reproduced with permission from [63]. Copyrights (2021), Elsevier.

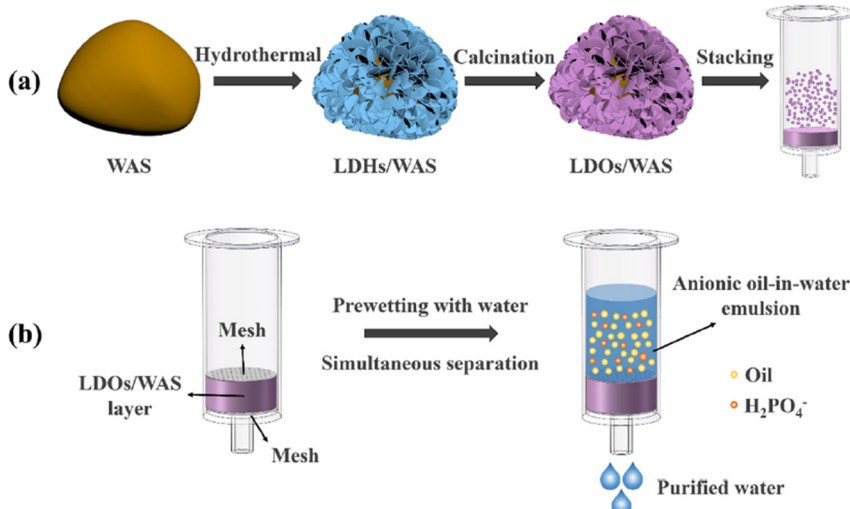

**Figure 4.** (**a**) Schematic preparation of aluminum sludge-based layer fabricated for oil water separation, (**b**) Underwater superoleophobic behavior for separation of oil-in-water emulsion. Reproduced with permission from [50]. Copyrights (2022), Elsevier.

The authors observed that the fabricated surface exhibited very high (>99.4%) separation ability for the oil/water emulsions using the gravity-directed method. The authors also observed that the developed layered double hydroxide/waste aluminum sludge was also efficient in the separation of surfactant-stabilized emulsions. Therefore, the author concluded that the environmental waste such as aluminum waste sludge has the ability to separate various oil/water emulsions.

Wu et al. [65] developed a polyvinylidene fluoride (PVDF) membrane with switchable wettability (underwater super oleophobic and under oil superhydrophobic) by the coating of waste pine wood along with sodium alginate via one step deposition process. In this work, initially, 0.1 g of sodium alginate was added in 10 mL of deionized water and allowed to be stirred for about 1 h until the formation of a uniform viscous solution. On

the other hand, authors utilized 0.1525 g of pine powders and dispersed it in 30 mL of deionized water and 3 mL of sodium alginate solution, stirring it for about 1 h. They used a vacuum pump for the fabrication of pine wood powder coated polyvinylidene fluoride membrane and dried it at room temperature. The switchable wetting behavior of the coated membranes was examined by the determination of the oil contact angle in underwater conditions as well as the water contact angle under oil conditions. Under water, they observed an oil contact angle of approximately 155° and under oil, a water contact angle of 157° (Figure 5) on the developed coated surface. Because of this special wetting behavior, the surfactant-stabilized oil/water emulsion separation ability of the developed coated PVDF membrane was examined via a filtration process. Authors observed excellent oil/water separation ability (>99%) even after 10 successive separation cycles with respect to various water and oil emulsions, including water-in-petroleum ether, water-in-diesel, water-in-kerosene, water-in-dichloromethane, water-in-heptane, water-in-dichloromethane and water-in-hexane. This behavior might be due to the gaps between the layers which are responsible for the formation of micro–nano scale channels. Thus, authors concluded that prepared waste pine wood coated membrane was highly effective for the oil/water emulsion separation even after the various repeating cycles.

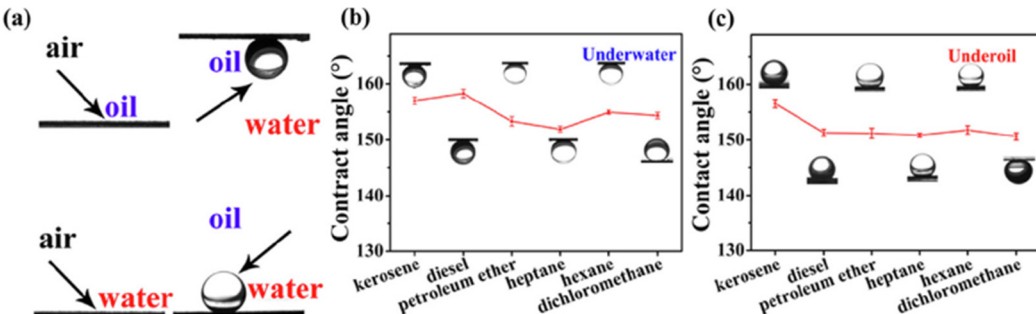

**Figure 5.** Schematic presentation of (**a**) switchable wettability of the developed membrane (underwater super oleophobic and under oil superhydrophobic), (**b**)Underwater oil contact angle exhibited by membrane, (**c**) Underoil water contact angle exhibited by membrane. Reproduced with permission from [65]. Copyrights (2020), Elsevier.

In another study, Li et al. [50] reported the fabrication of hierarchical ZnO nanopillars coating on the surface of waste brick grains (WBG) via an in situ growth process. The authors observed that the as prepared ZnO coated waste brick grains exhibited underwater superoleophobic (oil contact angle approximately 153°) properties. Thereafter, prepared ZnO/WBG material was further modified with a solution of vinyltriethoxysilane (1 wt %) in ethanol and water (*v/v*, 1:1). The modification of vinyltriethoxysilane led to the creation of a rougher surface which can induce superhydrophobic–superoleophilic properties (water contact angle = 147°) (Figure 6I).

The authors reported that gravity-driven separation efficiency of the fabricated ZnO/WBG surface for oil/water mixtures and oil-in-water and water-in-oil emulsions were >99% and >98%, respectively (Figure 6II). The coalescence of tiny droplets of water or oils of emulsions into bigger droplets on the ZnO/WBG surface was the major reason behind the efficacy of the developed materials in oil/water separation. Additionally, the authors believed that the prepared ZnO/WBG-based material with special wetting behavior is a low-cost set-up and has excellent separation efficiency along with the appreciable recyclability for the large-scale application.

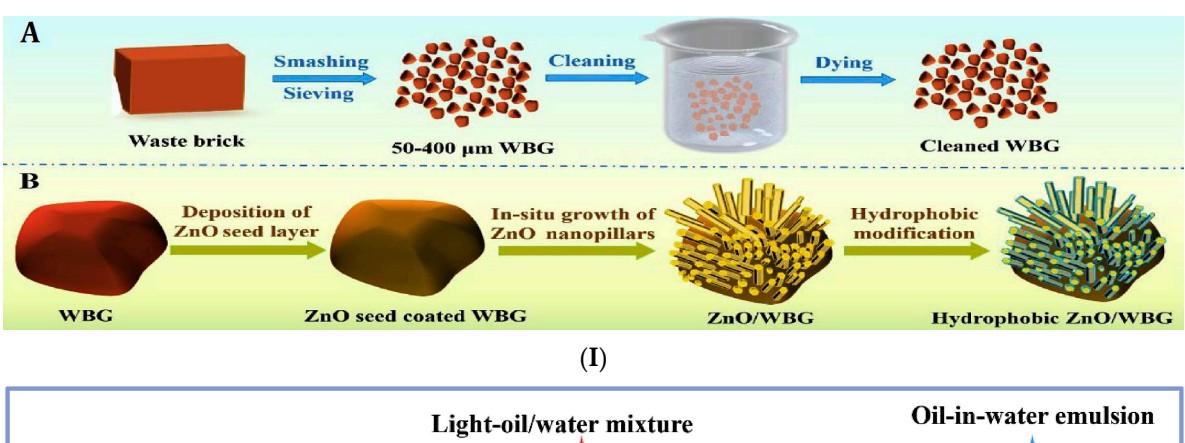

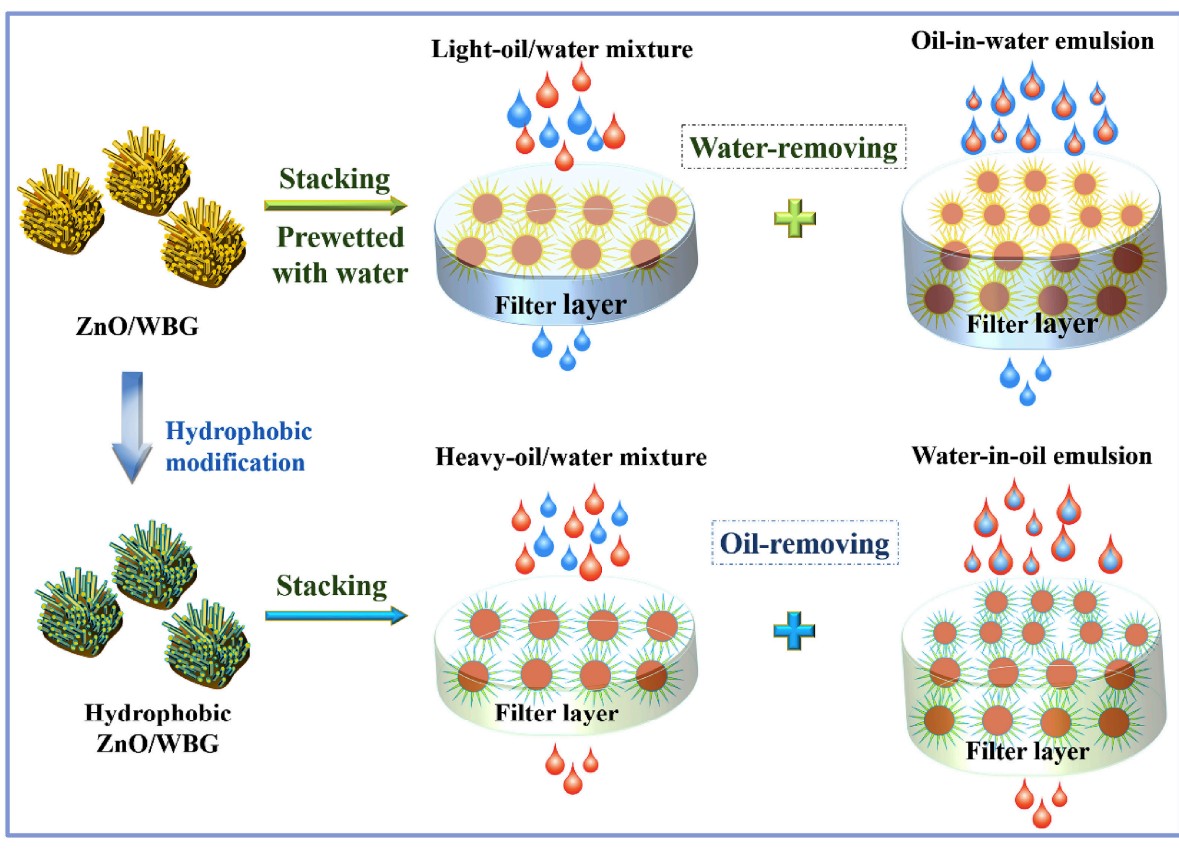

**Figure 6.** (**I**) Schematic illustration for the fabrication (**A**) waste brick grains from waste brick and (**B**) hydrophobic/superoleophilic ZnO/WBG materials by the use waste bricks grains using simple modification approach. Reproduced with permission from [50]. Copyrights (2020), Elsevier. (**II**) Schematic diagram of waste brick-based zinc oxide coated layer for oil/water separation. Reproduced with permission from [50]. Copyrights (2020), Elsevier.

### 2.2. Underoil Superhydrophobic-Superoleophilic Surfaces

Sun et al. [66] reported the fabrication of a grain-stalked filter layer with switchable wettability (under water superoleophobic and under oil superhydrophobic) using polyurethane plastic waste for oil/water separation. In this work, polyurethane plastic waste was crushed into grain of small sizes, washed with ethanol and dried in the oven at 60 °C for 4 h. Thus, obtained polyurethane plastic waste grains (PUPWG) were subjected to the coating of silica nanoparticles (SiO$_2$) via the dip-coating method using acidic silica sol in anhydrous ethanol and prepared SiO$_2$@PUPWG. Furthermore, silica nanoparticle coated PUPWG was again modified with a solution of octadecyltrimethylsiloxane (OTMS) dissolved in 200 mL of anhydrous ethanol via dip-coating approach along with sonication process for the formation of OTMS@SiO$_2$@PUPWG (Figure 7).

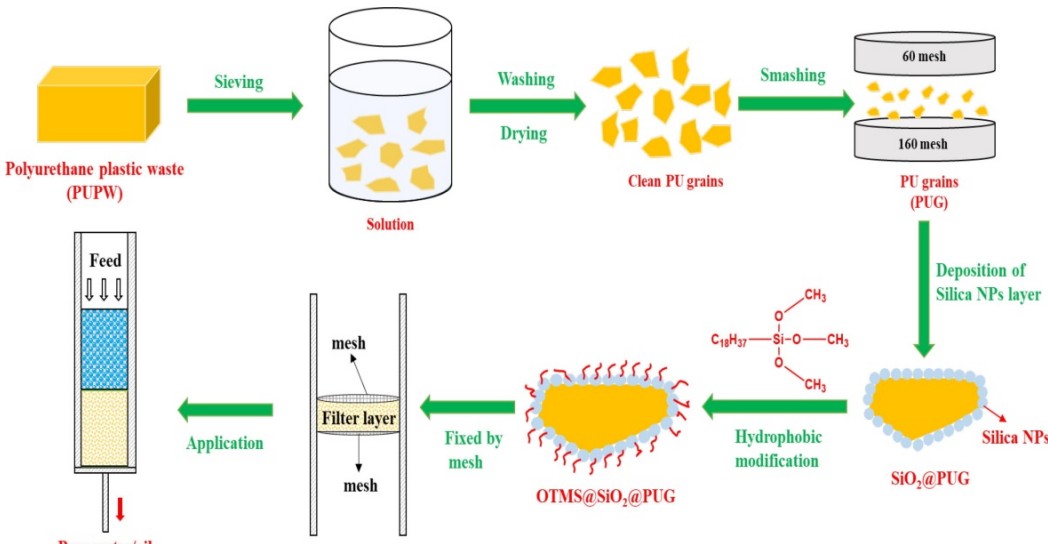

**Figure 7.** Schematic diagram of fabrication of waste polyurethane plastic-based mesh for oil water separation. Reproduced with permission from [66]. Copyrights (2021), Elsevier.

Interestingly, the authors observed that layer of SiO$_2$@PUPWG exhibits amphiphilicity within the air, however, in under water condition exhibited excellent oil repellency with an oil contact angle of 153.7°. In addition, the surface of OTMS@SiO$_2$@PUPWG exhibits superhydrophilicity in air with a water contact angle of 151.0°. This behavior might be due to the appearance of a hierarchical surface structure and lower surface energy.

In addition, the authors also observed that the developed coated OTMS@SiO$_2$@PUPWG surface exhibited excellent efficiency (>97%) to separate surfactant-stabilized emulsions of light oil (toluene) as well as heavy oil (dichloromethane) via a filtration approach under gravity with high flux as 679 L m$^{-2}$ h$^{-1}$. They reported that hydrophobicity and oil/water separation efficiency (97%) were maintained even after a 10 times recycling test under harsh conditions of temperatures 25–75 °C and pH = 1, 7 or 14.

Wang et al. [67] prepared a hydrophobic biomass-based aerogel from the waste corn stalks for the separation of a oil/water mixture via freeze-drying and dip-coating method (Figure 8). The authors developed porous three-dimensional aerogel with the help of konjac glucomannan (KGM), hexadecyltrimethoxysilane and corn stalk fibers (CSFs). In the synthesis of KGM/CSFs aerogel (KC), KGM and Na$_2$CO$_3$ were added into the distilled water and stirred for 1 h and then added 1% solution of corn stalks to the mixture and stirred it properly. In next step, the developed mixture was covered with plastic wrap and dried for about 1.5 h at 95 °C and then allowed to cool. After that, the prepared materials were frozen at −20 °C for 12 h and then again freeze-dried for 48 h for the formation of KGM/CSFs aerogel (KC). Thus, prepared KGM/CSFs aerogel was further modified with hexadecyltrimethoxysilane (>85%, HDTMS) solution (2.0% *v/v* in ethanol) via a facile dip-coating approach. This modification facilitates the formation of hydrophobic KGM/CSFs aerogel (HKC) (as shown in Figure 8I).

The wettability test results of modified hydrophobic KGM/CSFs aerogel (HKC) indicated the conversion of hydrophilic (water contact angle = 0°) surface to hydrophobic with a high water contact angle of 146°. This is dues to the lowering of surface energy along with formation of hierarchical structure on coated aerogel. Additionally, it was also observed that modified aerogel (HKC) floated on the water surface because of its low density and excellent water repellency/hydrophobicity [Figure 8II].

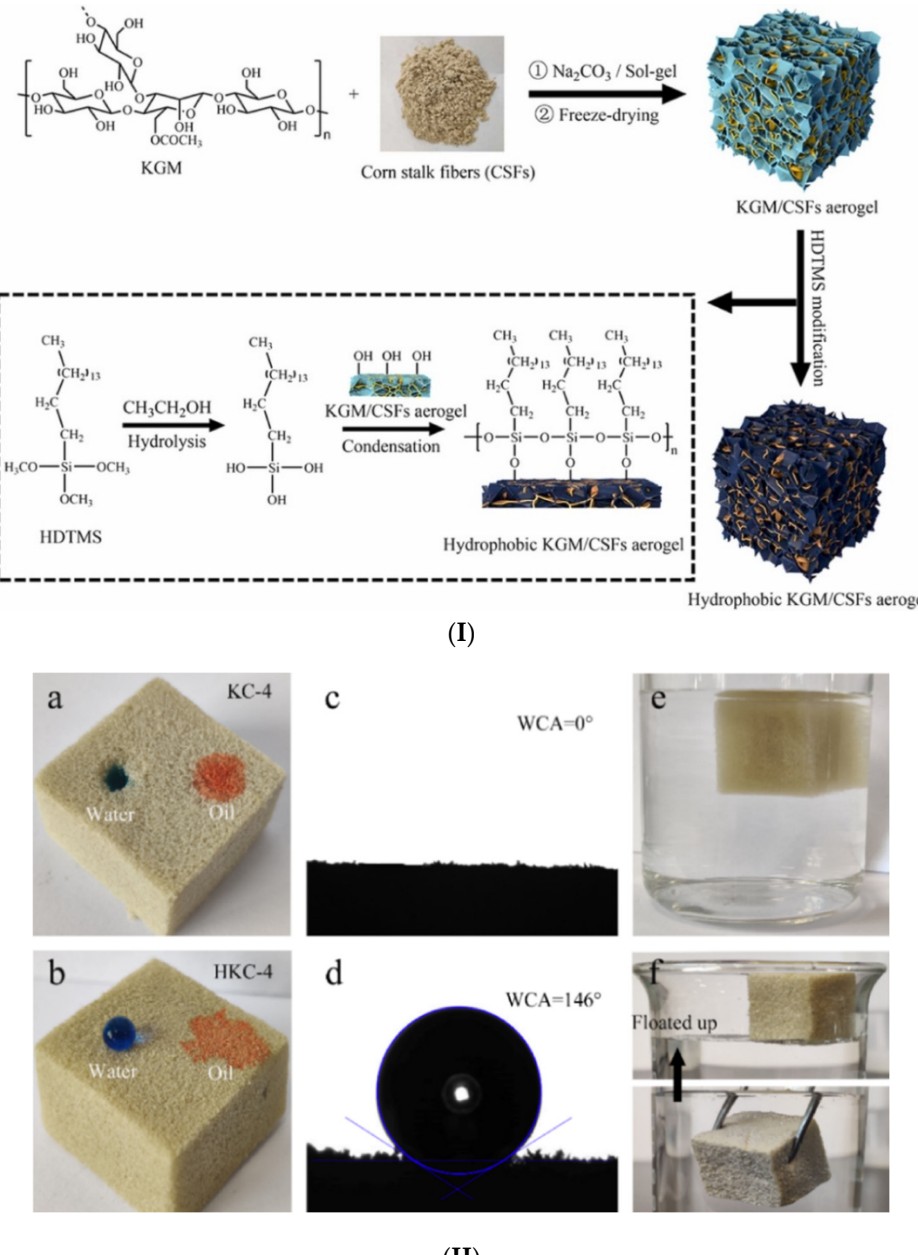

**Figure 8.** (**I**) Schematic diagram of fabrication of corn-stalks based hydrophobic aerogel membrane for oil water separation. Reproduced with permission from [67]. Copyrights (2022), Elsevier. (**II**) The wetting behavior of water and oil (toluene) on the surface of KC (**a**) and HKC (**b**); The value of water contact angle on the surface of KC (**c**) and HKC (**d**); Hydrophilic behavior of KC (**e**) and hydrophobicity (floating ability on water surface) of HKC (**f**) during immersion in water. Reproduced with permission from [67]. Copyrights (2022), Elsevier.

Thus, the as-prepared modified KGM/CSFs aerogel was useful for the oil/water separation application because of their selective wetting behavior. The results of the oil/water separation study imply that the modified KGM/CSFs aerogel exhibited high absorption capacity in the range 24.76–58.07 g g$^{-1}$ for various oils including soybean oil, diesel oil, toluene, hexane, kerosene, petroleum ether, dichloromethane and chloroform. In addition, it was also observed that oil/organic solvents absorption efficacy of the modified aerogel was maintained even after 10 recycling tests. Thus, the authors concluded that waste corn stalk fiber-based hydrophobic KGM/CSFs aerogel can be used for large-scale application because of its great potential and simplicity of fabrication process.

Yu et al. [68] fabricated a micrometer sized porous material by one-step derivatization method using kraft lignin which is a paper and pulp industry waste. In this work, initially, imidazole (3.5 g) was added in the aqueous solution of kraft lignin (2 g/60 mL of water) under inert atmosphere and stirred at room temperature for 1 h. After, trichloro(octyl)silane (8 mL) was added to the resultant stirred mixture of lignin and imidazole, performing a reaction at 50 °C for 24 h under a nitrogen atmosphere in order to obtain silane-coated lignin particles. The authors observed that the silane-coated lignin particles exhibit excellent water repellency (superhydrophobicity) with a water contact angle of 156°.

The oil/water separation efficiency of the developed coated lignin particles was examined with respect to simple oil/water mixtures, surfactant-stabilized water/oil emulsions (tween80-stabilized chloroform-in-water emulsion) by using modified lignin particles as a filter or as clump (as shown in Figure 9). They reported excellent efficiency (>98%) of the silane–lignin filter for a variety of oil/water mixtures (i.e., diesel, gasoline, vegetable oil, dodecane, chloroform) including tween80-stabilized chloroform-in-water emulsion.

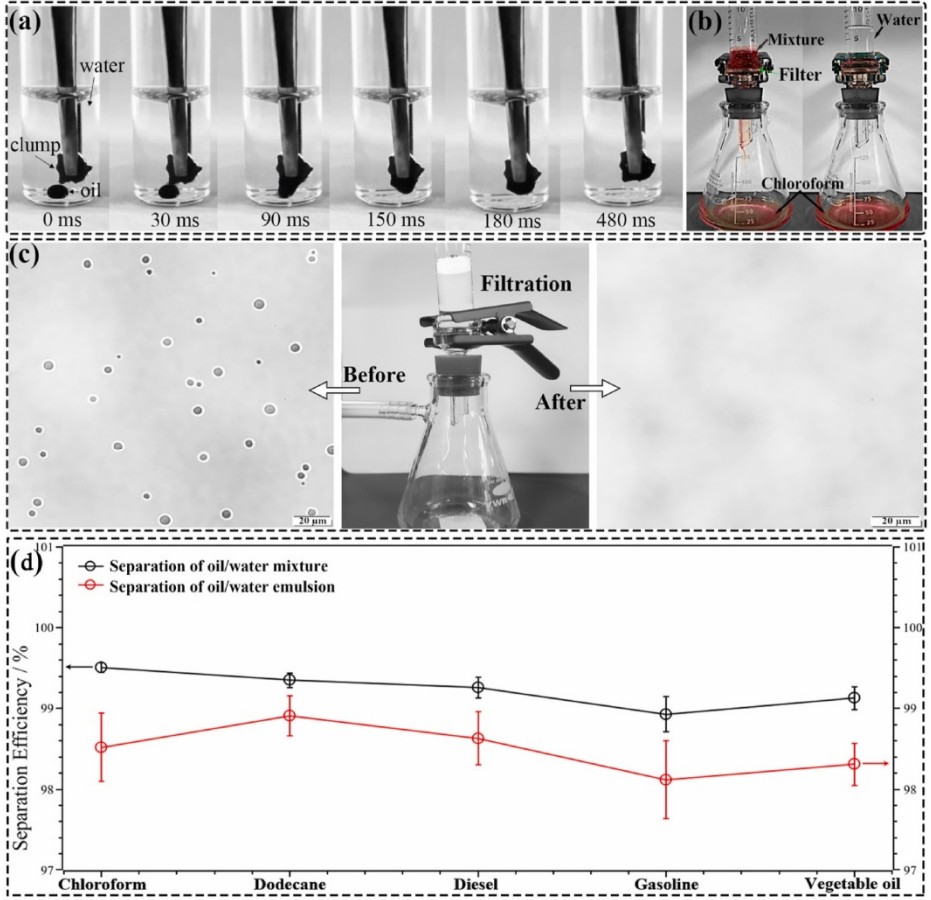

**Figure 9.** (**a**) The process of oil/water mixtures using silane–lignin clump, (**b**) separation of oil/water mixtures via filtration, (**c**) images of emulsion before and after separation via filtration assembly using silane–lignin particles, (**d**) silane–lignin separation efficiency for separation of various oil/water mixtures as well as emulsions. Reproduced with permission from [68]. Copyrights (2020), Elsevier.

Zhang et al. [69] fabricated a superhydrophobic surface using dodecanethiol-modified polypyrrole (Ppy) particles coating cigarette filters (solid waste) by a dip-coating method since polymerization of silane coupling agents is the common method for the formation of hierarchical micro/nanostructures and superhydrophobicity [70,71]. In this study, polymerization of pyrrole (with different concentrations) was chosen to be the compatible with the basic components (cellulose acetate) of cigarette filters for inducing hierarchical micro/nanostructures. A water contact angle of 150° with a sliding angle of 10° was mea-

sured on the surface of dodecanethiol-modified Ppy (1 mol L$^{-1}$) cigarette filters (Figure 10). The improved water contact angle on the coated cigarette filter was due to the formation of micro–nano hierarchical structures and lowering the surface energy of the modified surfaces with the increase in pyrrole concentration.

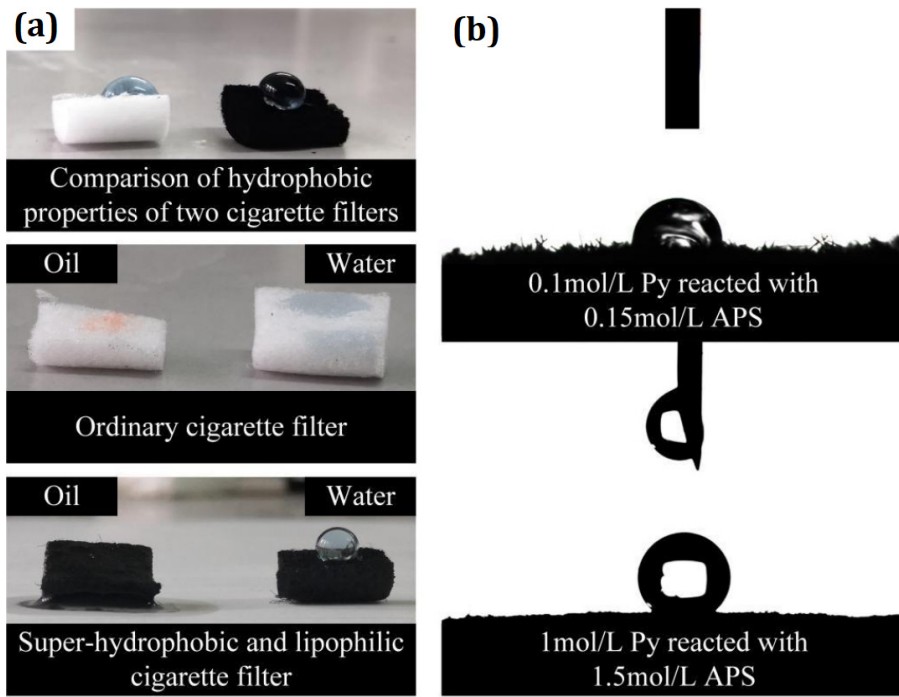

**Figure 10.** (**a**) Wetting behavior of unmodified cigarette filter and dodecanethiol-modified Ppy cigarette filters, (**b**) effect of Ppy concentration on water contact angle value for the modified cigarette filters. Reproduced with permissions from [69]. Copyrights (2020), MDPI.

The authors employed the developed dodecanethiol-modified Ppy cigarette filters to separate various types of oil/water mixtures, including petroleum ether/water, engine oil/water, chloroform/water and they observed high separation efficiency (>96%) in each oil/water mixture even after 30 separation cycles. Thus, on the basis of experimental findings, the authors concluded the development of oil/water separation filter using waste cigarette filters to not only be useful in cleaning up oil spills but also in being applied to reduce solid waste.

In addition, other waste materials-based surfaces with special wetting behavior (underwater superoleophobic-hydrophilic and underoil superhydrophobic–oleophilic) that have been examined recently in many studies for oil/water separation application either via filtration or the adsorption process are summarized in Table 1. In summary, the above literature findings show that waste material-derived surfaces with special wetting behavior, including agricultural, industrial, cellulose, plastic, bricks, fruit peels and fruit shells are efficient and exhibit remarkable separation efficiency for various types of oil/water mixtures. However, most of the studies are not reported on the development of such surfaces for large-scale production and its application is still not examined in real environmental conditions at commercial scale. In future studies, these issues need to be addressed for the development of efficient, economic and sustainable solutions for oil/water separation including emulsions.

**Table 1.** Waste material-based underwater superoleophobic–superhydrophilic and underoil superhydrophobic–superoleophilic surfaces for oil/water separation.

| Mechanism | Separation Process | Waste Material | Sources of Waste Materials | Other Material for Surface Modification | Coating Method | Contact Angle | Separation Efficacy | Flux/Sorption Capacity | Separated Oils | Comments | Ref. |
|---|---|---|---|---|---|---|---|---|---|---|---|
| Hydrophobic /oleophilic | Sorption | Waste tissue paper | Cellulose waste | Polyvinyl alcohol, sodium chlorite, sodium hydroxide, tetraethyl orthosilicate | Freeze-drying | WCA 154.93° ± 4.14 | - | 69−168 g g$^{-1}$ Sorption capacity | Chloroform, motor oil, acetone, DMF, olive oil and toluene | The modified aerogel exhibits sorption efficay greater than 92% even after repeating 20 cycles and 89% up 50 cycles of repeation process. | [72] |
| | | Egg carton material | Cellulose waste | Acetone, candle soot | Dip-coating | WCA = 142.2°, OCA = 43.1° | - | Absorption: 3 g g$^{-1}$ | Petrol, diesel, refined oil, coconut, engine and mustard oil | The maximum and minium sorption capacity of the modified surfaces were 3.1 and 1.6 g g$^{-1}$ for mustard oil and petrol, respectively. | [73] |
| | | Waste bamboo | Agricultural waste | Methyl trimethoxy silane, Anhydrous Alcohol | Chemical vapor deposition and delignification | WCA = 153°, OCA = 0° | - | Absorption: 18.8 g g$^{-1}$ | Silicone oil, diesel oil, chloroform, paraffin oil and toluene | The highest oil absorption capacities of the developed surfaces was >40–55%. | [74] |
| | | Epoxy resins | Industrial waste | Epoxy oligomer, 4, 4-diaminodiphenyl methane, N-methyl kelopyrrolidide | Dip-coating | WCA = 146.5° | - | Absorption: 116 g g$^{-1}$. | Soybean oil, colza oil, pump oil, n-hexane, chloroform and silicone oil | The selective oil absorption ability of developed surface stabilized was approximately 53 g g$^{-1}$ with respect to gasoline even after the 10 successive repetition of separation cycles. | [75] |
| | | Skin collagen fiber waste | Industrial waste | Methacrylic acid, glycidyl Methacrylate, sodium dodecyl sulphate, dodecyl Mercaptan | In situ free radical poly-merization | WCA 145° | 99.93 ± 0.03% | – | Soybean oil and motor oil | The developed modified surfcace exhbited high separation efficiency (>99.93%) even after 10 cycles and 3 cycles for surfactant free oil in water emulsion and surfactant-stabilized oil-in-water emulsion respectively. | [76] |

**Table 1.** *Cont.*

| Mechanism | Separation Process | Waste Material | Sources of Waste Materials | Other Material for Surface Modification | Coating Method | Contact Angle | Separation Efficacy | Flux/Sorption Capacity | Separated Oils | Comments | Ref. |
|---|---|---|---|---|---|---|---|---|---|---|---|
| | | Polystyrene foam | Plastic waste | Styrene, Tetraethyl orthosilicate, Ammonia solution and hexade-cyltrimethoxysi-lane | Pickering emulsion (HIPPE) technique | - | - | 20.4–58.1 g g$^{-1}$ Absorption capacity | Dichloromethane, chloroform, acetone, hexane, dichloromethane, acetic ether, methanol, ethanol, toluene, peanut oil, diesel, pump oil and crude oil | The fabricated foam adsorb oil and revealed the adsorption capacity above 90% even after the 10 consecutive cycles. | [77] |
| | | Waste polystyrene | Plastic waste | Ethyl acetate, craft polystyrene | Blow spinning | WCA 138° OCA 0° | 97% | - | Diesel oil | After repeating consecutive two cycles the separation efficacy of the modified material was >90%. | [78] |
| | Filtration | Thermosetting resins | Plastic waste | - | Simple Mechanical crushing | WCA 117° | 97% | 15 987 L m$^{-2}$ h$^{-1}$ s Separation flux | Toluene, chloroform, n-hexane and gasoline | The prepared surface can easily separate the oil/water mixtures as well as emulsion with the droplet size more than 50 nm. | [79] |
| | | Sugarcane bagasse ash | Agricultural waste | Methyl triethoxysilane, tetraethoxysi-lane | Sol–gel process | WCA 163.9° | 99.9% | Oil flux: 137.2 L m$^{-2}$ h$^{-1}$ s | Crude oil | The separation efficacy of the modified surfaces was examined in consideration of process variables such as grafting time (30 to 60 min), grafting cycle (1–4 cycles), and calcination temperature (400–600 °C) to separation effieciency. | [80] |

Table 1. *Cont.*

| Mechanism | Separation Process | Waste Material | Sources of Waste Materials | Other Material for Surface Modification | Coating Method | Contact Angle | Separation Efficacy | Flux/Sorption Capacity | Separated Oils | Comments | Ref. |
|---|---|---|---|---|---|---|---|---|---|---|---|
| Underwater super-oleophobic/hydrophilic | Filtration | Polyurethane foams | Plastic waste | Ferric chloride, dopamine, ammonium hydroxide, polydopamine | Polymerization | Underwater OCA 145.7 ± 2.8 | 98.7% | Water flux higher than 57,796 L $h^{-1}$ $m^{-2}$ | Hexane, cyclohexane, liquid paraffin, pump oil and petroleum ether | The oil/water separation effucacy of the developed surface of modfied foam was >97% even after the 100 cycles of the recycling test. | [81] |
| | | Carbon fibers | Industrial waste | Cellulose filter papers, tannic acid, tris(hydroxymethyl)- Aminomethane (Tris), (3-Aminopropyl) triethoxysilane, sodium Dodecyl sulphate, sodium hydroxide, hydrochloric Acid, dichloromethane (DCM) | Pyrolysis method | Underwater OCA 157.2° | 99.8% | - | Dichloromethane (DCM), canola oil, n-hexane, kerosene and silicone oils | The developed surface has high roughness 2.1 times higher than the raw material as well as effective surface area was 1.6 times higher than the control due to development of micro–nanostructured sructured on the surface. | [49] |
| | | Coal fly ash | Industrial waste | Coal fly ash, distilled water | - | Underwater OCA 155 ± 2° | 99.9% | - | Kerosene, diesel and hexane | Oil/water separation efficacy was directly proportional to the thickness of the separating membrane. However separating flux was decreases from 1050 L $m^{-2}$ $h^{-1}$ to 513 L $m^{-2}$ $h^{-1}$ with the increase of thickness. | [82] |

**Table 1.** *Cont.*

| Mechanism | Separation Process | Waste Material | Sources of Waste Materials | Other Material for Surface Modification | Coating Method | Contact Angle | Separation Efficacy | Flux/Sorption Capacity | Separated Oils | Comments | Ref. |
|---|---|---|---|---|---|---|---|---|---|---|---|
| | | Natural shell | Raw materials discarded on the beach | Perfluorooctanoic acid, (3-aminopropyl) triethoxysilane and bis (3-(trimethoxy silyl) propyl) amine, Spray-Mount™ Super 75 | Dip-coating method | Underwater OCA 154° | 99.3% | - | Chloroform, olive oil, decane and hexadecane | The developed surface was capable to separate both oil/water and oil/oil mixtures of different polarity. | [83] |
| | | Pomelo Peel fibers | Fruit waste | Anhydrous ethanol, waterborne polyurethane | Spraying method | Underwater OCA | 97.0% | - | Kerosene, motor oil and soybean oil, hexane, liquid paraffin and chloroform | The modified mesh exhbited high separation efficiency (>97.0%) in oil/water mixtures of different pHs. | [84] |
| | | Coconut shell waste | Fruit waste | Quartz sand, waterborne polyurethane | Dip-coating method | Underwater OCA 151.2° | 99.92% | - | Hexane, dichloromethane, trichloromethane, anhydrous ethanol and petroleum ether | The developed material exhibits the permeability coefficient of organic solvents such as cyclohexane, petroleum ether, dichloromethane, and trichloromethane were higher than 10 m/h than water 9.37 m/h. | [85] |
| | | Waste cigarette filter | Cellulose waste | Trichloromethane, acetone and N, N-dimethylformamide | Electrospinning approach | Underwater OCA | 99.9% | Water flux was about 1000 L m$^{-2}$ h$^{-1}$ | Kerosene, diesel, petroleum ether, hexane and trichloromethane | The prepared surface exhibits underwater superoleophobicity and underoil super hydrophobicity. In addition, separation efficiency was >99% even after the 10 repeating cycles of sepration process. | [62] |

**Table 1.** *Cont.*

| Mechanism | Separation Process | Waste Material | Sources of Waste Materials | Other Material for Surface Modification | Coating Method | Contact Angle | Separation Efficacy | Flux/Sorption Capacity | Separated Oils | Comments | Ref. |
|---|---|---|---|---|---|---|---|---|---|---|---|
| | | Waste Paper | Cellulose waste | - | - | Underwater OCA 151 $\pm$ 2.5° | 99% | - | Soyabean oil, cyclohexane and hexane | The modified waste papers exhbited very flux of oil/water mixture sepration revealed water flux, i.e., 1126, 1837, 3246, 1273, and 1145 $L \cdot m^{-2} \cdot h^{-1}$ for stickers, notebook, lens paper, envelopes, and receipts, respectively. | [86] |
| | | Bricks Powder | Waste bricks | Distilled water and ethanol | Physical refining process | Underwater OCA 150° | 98.4% | Flux 4384 L $m^{-2}$ $h^{-1}$ | Hexane, methylbenzene, trichloromethane and absolute ethyl alcohol and soyabean oil | The fabricated waste bricks granules (100–400 μm) have superhydrophilicity and superoleophilicity in air and under-liquid amphiphobic properties. | [87] |

### 3. Summary and Future Work

In the field of oil/water separation, a large part of the materials with special wetting behavior utilized fluorine-based chemical modifications for the desired surface characteristics. However, fluorinated compounds are not only hazardous to the environments and human health but also expensive. Therefore, scientists have changed their attention to readily available waste materials (including waste bricks, fruit peel and fruit shell, waste agricultural by-products and cigarette filters) that are inherently hydrophilic and easily processed to develop surfaces with special wetting behavior. It is reported that waste materials based surfaces with special wetting behavior exhibited high efficiency in the separation of surfactant-stabilized emulsions as well as normal oils or organic solvent mixtures with water. In this review, oil/water separation with waste materials modified by different methods are described along with the possible mechanism. From the literature survey, it was observed that by the adjustment of surface roughness and surface energy or underwater oleophobicity and underoil hydrophobicity, these waste materials can efficiently work as a filter or adsorbent for the separation of both emulsified oil/water mixtures as well as immiscible oil/water mixtures. Despite the great potential and high efficacy of special wettability waste materials for oil/water separation, there are some significant challenges during their application in real-field contaminated water.

(1) Most of the studies are still limited to the lab scale only, and are very difficult to apply at the industrial scale. Therefore, it is highly essential to examine their applicability in real case problems with the oil of different viscosities under natural environmental conditions.

(2) The matrix of oily contaminated wastewater is more complex. Therefore, design and development of multi-functional materials should be focused on performing real field applications.

(3) Generally, special wettability waste materials have weak mechanical properties and a very small service life. However, longevity is one of the essential factors during its practical application to treat real field contaminated water at a large scale. Therefore, further research must be focused on the combination of theoretical and experimental investigations to achieve industrialization and large-scale oil/water separation in the real field practical application of special wettability waste materials with a continuous mode of operation.

**Funding:** This research received no external funding.

**Data Availability Statement:** All data generated or analyzed during this review study are included in this submitted manuscript and their respective references are provided in the reference list.

**Acknowledgments:** The author acknowledges the support from the Department of Chemistry, and Research & Development Cell of Maharishi Markandeshwar (Deemed to be University), Mullana, Ambala, Haryana, India.

**Conflicts of Interest:** The author declares no conflict of interest. The funder had no role in the design of the study; in the collection, analyses, or interpretation of data; in the writing of the manuscript; or in the decision to publish the results.

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
