# Peer review of "Oil/Water Separation Using Waste-Derived Functional Materials with Special Wetting Behavior"

_resources, doi:10.3390/resources11100083_

Round 1

Reviewer 1 Report

1.       Author introduces an interesting topic in oil/water separation using waste which achieves SDGs goals of the UN. Based on that please mention about the importance of your work globally in terms of using waste materials and achieving SDGs in one paragraph.

2.       From section 2.1 each paragraph is devoted to describe only one figure, please enrich your text with other peer review articles and examples even you will not insert images.

3.       I have notice that more than 50% of references are in introduction section which not logic especially in review paper. How about discussion in the following sections?

4.       In table.1, add one column about the source of waste materials from industry or environment according to figure.1.

5.       In page .13: give more details and discussion about table.1 and its important in this review.

6.       In conclusion: “In this review, oil/water separation with waste materials modified by different methods are described along with the possible mechanism “which mechanism described in the text. c

7.        The following most related pear review articles are recommended to strength the introduction, other section in the review. Colloids and Surfaces A: Physicochemical and Engineering Aspects, Volume 651, 20 October 2022, 129655, “Nanocellulose Membranes for Water/Oil Separation” , Handbook of Nanocelluloses pp 1–37, Membranes for Oil/Water Separation: A Review, advanced materials interface, Materials Letters, Volume 306, 1 January 2022, 130965 , Polymers ,MDPI, 2020, 12(11), 2597; Environmental Nanotechnology, Monitoring & Management, Volume 14, December 2020, 100314

Reviewer 2 Report

This review discussed oil/water separation application by water materials-based special wettable surfaces and the mechanisms of different methods have been discussed. I think authors have made a good word and this paper can be considered for publication in Resources. Some suggestions and questions are listed as follow,

(1) In Figure 8, can authors add more introduction for different pictures?

(2) In this review, authors reported methods from different paper. They only simply introduced their work but with little discussion and comparison. They lack their own point of view.

(3) The summary of this review is too simple. Authors can add more discussions and conclusions.

Round 2

Reviewer 1 Report

 Comments covered from Authors